# No Excess of Mortality from Lung Cancer during the COVID-19 Pandemic in an Area at Environmental Risk: Results of an Explorative Analysis

**DOI:** 10.3390/ijerph20085522

**Published:** 2023-04-14

**Authors:** Francesco Addabbo, Massimo Giotta, Antonia Mincuzzi, Aldo Sante Minerba, Rosa Prato, Francesca Fortunato, Nicola Bartolomeo, Paolo Trerotoli

**Affiliations:** 1School of Medical Statistics and Biometry, University of Bari Aldo Moro, Azienda Sanitaria Locale Taranto, 74121 Taranto, Italy; francescodr.addabbo@gmail.com; 2School of Medical Statistics and Biometry, Department of Interdisciplinary Medicine, University of Bari Aldo Moro, 70124 Bari, Italy; massimo.giotta@uniba.it; 3Unit of Statistics and Epidemiology, Azienda Sanitaria Locale Taranto, 74121 Taranto, Italy; 4Hygiene Unit, Policlinico Riuniti Foggia Hospital, Department of Medical and Surgical Sciences, University of Foggia, 71122 Foggia, Italy; 5Department of Interdisciplinary Medicine, University of Bari Aldo Moro, 70124 Bari, Italy; paolo.trerotoli@uniba.it

**Keywords:** excess mortality, COVID-19, environmental risk area, forecasting model, lung cancer

## Abstract

Background: The COVID-19 pandemic and the restrictive measures associated with it placed enormous pressure on health facilities and may have caused delays in the treatment of other diseases, leading to increases in mortality compared to the expected rates. Areas with high levels of air pollution already have a high risk of death from cancer, so we aimed to evaluate the possible indirect effects of the pandemic on mortality from lung cancer compared to the pre-pandemic period in the province of Taranto, a polluted site of national interest for environmental risk in the south of Italy. Methods: We carried out a retrospective observational study on lung cancer data (ICD-10: C34) from the Registry of Mortality (ReMo) for municipalities in Taranto Province over the period of 1 January 2011 to 31 December 2021. Seasonal exponential smoothing, Holt–Winters additive, Holt–Winters multiplicative, and auto-regressive integrated moving average (ARIMA) models were used to forecast the number of deaths during the pandemic period. Data were standardized by sex and age via an indirect method and shown as monthly mortality rates (MRs), standardized mortality ratios (SMRs), and adjusted mortality rates (AMRs). Results: In Taranto Province, 3108 deaths from lung cancer were recorded between 2011 and 2021. In the province of Taranto, almost all of the adjusted monthly mortality rates during the pandemic were within the confidence interval of the predicted rates, with the exception of significant excesses in March (+1.82, 95% CI 0.11–3.08) and August 2020 (+2.09, 95% CI 0.20–3.44). In the municipality of Taranto, the only significant excess rate was in August 2020 (+3.51, 95% CI 0.33–6.69). However, in total, in 2020 and 2021, the excess deaths from lung cancer were not significant both for the province of Taranto (+30 (95% CI −77; +106) for 2020 and +28 (95% CI −130; +133) for 2021) and for the municipality of Taranto alone (+14 (95% CI −47; +74) for 2020 and −2 (95% CI −86; +76) for 2021). Conclusions: This study shows that there was no excess mortality from lung cancer as a result of the COVID-19 pandemic in the province of Taranto. The strategies applied by the local oncological services during the pandemic were probably effective in minimizing the possible interruption of cancer treatment. Strategies for accessing care in future health emergencies should take into account the results of continuous monitoring of disease trends.

## 1. Introduction

The restrictive measures adopted to reduce the impact of the COVID-19 pandemic in health facilities resulted in delays in the treatment of other diseases, such as cancer, with a possible increase in mortality compared to the expected rate [1]. In the province of Taranto, which is already known for having environmental risks related to the presence of one of the largest steel plants in the EU, the risk of death from cancer has been shown to be the highest in the region—especially for lung cancer [2].

Taranto is a city in southeastern Italy with a high population density, and it suffers from serious environmental pollution as a result of well-established industrial plants such as a major oil refinery, a concrete factory, a naval military base, and one of the largest integrated cycle steel foundries in Europe. In 2002, the Italian government included Taranto in the list of polluted sites of national interest in order to implement environmental remediation [3]. 

Early in 1997, the WHO carried out the first epidemiological study based on an environmental design aimed at evaluating the health status of the population living in Taranto [4]. Several studies followed that report, most of which showed rates of all-cause and all-cancer mortality higher than the regional mortality rates. Significant excesses were observed for cancer mortality, and significant results were displayed for lung, pleural, bladder, liver, and lymphohematopoietic cancers [2,5,6,7,8]. Over the past 20 years, all of this has greatly sensitized public opinion and awareness, leading in a few cases to public riots and social instability among the residents. Significant financial and human resources have been employed in efforts to deal with this emergency health issue, which appeared to be linked to the need to live in an area that was simultaneously economically developed and environmentally safe.

In this scenario, the current outbreak of SARS-CoV-2 (the causative agent of COVID-19) led to a significant worsening of several public health issues. Between March 2020 and December 2021, in correspondence with the epidemic waves, different levels of lockdown were adopted in Italy. In the first phase of the spread of SARS-CoV-2, a total lockdown was adopted with a ban on leaving the house except out of necessity, suspension of educational services, and closure of all commercial activities and public offices [9]. Subsequently, in the periods with moderate levels of restrictions, there was a ban on moving out of town except for work, suspension of educational services, and closure of some commercial activities [10,11]. In the period with a very low incidence of SARS-CoV-2, there were few restrictions: suspension of educational indoor services and a reduction in the number of people accessing commercial activities [12,13]. The consequences of the strict measures implemented in order to reduce the burden of the spread of the virus were felt in many aspects of life—especially in terms of public health. During the first months of the outbreak, the Italian Regional Health Systems diverted nearly all of their prevention and healthcare treatment settings toward emergency care in an attempt to reduce the spread of the virus and to cope with the influx of COVID-19 patients [14,15].

Pandemic health policies have had a tremendous impact on patients with cancer. Early in the pandemic, a WHO survey found that health services for non-communicable diseases in many countries were partially or completely disrupted, which was later confirmed and systematically reviewed in several studies [16,17,18]. In the Province of Taranto, due to the restrictions imposed during the first year of the pandemic [9,10], there was a reduction, compared to 2019, in access to medical check-ups in pulmonology (−72%), radiotherapy (−46%), and oncology (−35%).

Patients with cancers, particularly those with lung cancers, have been reported to have more severe outcomes from coronavirus disease 2019 (COVID-19), including higher rates of hospitalization and death [19,20,21,22]. There is a debate as to whether lung cancer itself or other pre-existing factors, such as significant cardiovascular and respiratory comorbidities, older age, genetic variation in immunity, smoking-related lung damage, underlying cardiopulmonary disease, and/or cancer-directed treatments, predispose individuals to significant symptoms of severe acute respiratory syndrome coronavirus 2 (SARS-CoV-2) infection [22,23]. Lung cancer represents a central challenge in clinical diagnosis and treatment decision making in the context of the highly contagious COVID-19 pandemic. 

The natural history of this disease has significantly changed in recent years thanks to the development of novel medical treatments and less-invasive surgical procedures. Chemotherapy, radiation, and surgical therapies are keystones in the treatment of early stage and locally advanced lung cancer, with good prognosis. Chemotherapy plus targeted therapy or immunotherapy has also brought substantial survival benefits to patients with recurrent/metastatic advanced cancer [24]. A recent report from the Memorial Sloan Kettering Cancer Center by Himmelman et al. evaluated the impact of immunotherapy in lung cancer patients affected by COVID-19, investigating whether programmed death-1 (PD-1) blockade affects the severity of COVID-19 in patients with lung cancer. The results, based on 69 consecutive patients with lung cancer, showed that treatment with an anti-PD-1 monoclonal antibody was not associated with an increased severity of COVID-19 [25].

However, frequent outpatient medical consultations or follow-up imaging in hospital facilities during the COVID-19 pandemic, together with receiving immunosuppressive anticancer treatments, might have considerably increased the risk of becoming infected. For oncologists tasked with the care of this particular patient population, deciding whether to offer, postpone, or even cancel treatments has become a crucial recurring dilemma, with the continuous need to balance the benefits of in-hospital anticancer treatments against the risk of infection. Even for cancer patients who are not affected by COVID-19, the disruption of regular hospital services, such as imaging and treatment appointments, can have a significant impact on the physical and mental wellbeing of these patients [15]. While the health service is now beginning to return to normality—or to the “new normality”—the knock-on consequences in cancer care and the outcomes of patients with lung cancer will definitely be felt for years to come. The pandemic has had a profound impact on all aspects of the lung cancer pathway, starting from delayed patient presentation, through to diagnosis, treatment, and follow-up. Preliminary raw data analyses from our group in Taranto’s Local Health Authority (unpublished data) have suggested that patients’ visits to treatment facilities have been curtailed and treatments have, in some cases, been withheld, and there is also evidence of a negative impact on cancer-related quality of life, as also extensively demonstrated by other authors [26].

In this retrospective study, we aimed to analyze whether the measure of restrictions on health services during the COVID-19 pandemic caused an excess of mortality from lung cancer in the area of Apulia at a high environmental risk. This study can therefore offer a comprehensive overview of the impact of the COVID-19 pandemic on the mortality from bronchial and lung cancers in a polluted area. In this contribution, we performed an analysis of the excess mortality from lung cancer during the first 18 months of the SARS-CoV-2 outbreak across Taranto Province.

## 2. Materials and Methods

### 2.1. Data

We conducted a retrospective observational study using anonymous aggregated data extracted from the administrative healthcare databases of the Local Health Authority (LHA) site in Taranto Province. Data were collected and stored in the Regional Information System. Access to data is regulated by a regional policy to allow the use of data for epidemiological purposes by the Taranto LHA’s Unit of Epidemiology and Statistics. Data were provided after anonymization and treated according to the current national laws for treatment of health data. We analyzed the Registry of Mortality (ReMo) for deaths related to malignant neoplasms of the bronchi and lungs (ICD-10 C34) among residents from the province of Taranto from 1 January 2011 to 31 December 2021.

Monthly data aggregated by gender and age group on lung cancer deaths recorded in Apulia between 2015 and 2019 were extracted from the databases available via the Regional Epidemiological Observatory.

Data on the number of residents in the municipality of Taranto, the province of Taranto, and the entire region of Apulia were collected from the Demo ISTAT [27] and stratified by sex and age class (<65 years, 66–75 years, 76–85 years, or 86+ years). The crude and adjusted mortality rates per 100,000 inhabitants (MRs) were determined as the ratios between new cases and the population multiplied by 100,000.

### 2.2. Standardization

The 2011–2021 series of monthly lung cancer mortality rates in the province of Taranto and for the municipality of Taranto were standardized by sex and age class via an indirect method. The average monthly lung cancer mortality rates for the five-year period of 2015–2019, stratified by gender and age classes and calculated on the entire Apulian population, were used as a reference to calculate the expected deaths. The monthly standardized mortality ratio (SMR) was calculated as the ratio between observed deaths and expected deaths, while the adjusted mortality rate (AMR) was calculated as a product between the SMR and the Apulian monthly average mortality rates.

### 2.3. Forecasting Models

The excess mortality due to specific circumstances is not directly observable, because it would be necessary to know how many events would occur outside those circumstances. Therefore, we attempted to estimate the excess mortality using forecasting models [28]. These models used historical data as a reference and, subsequently, allowed us to compute the mortality trends from the data. In our study, the COVID-19 pandemic was the particular circumstance, so the period of this study was split into two parts: a training set (from 1 January 2011 to 28 February 2020) and a forecasting set (from March 2020 to December 2021). The excess mortality rate was calculated as the difference between the expected AMR estimated without the pandemic from March 2020 to December 2021 and the AMR that was observed in the same period. The excess mortality rates were used to compute the number of COVID-19-related excess deaths by multiplying the predicted excess mortality rate by person-years of exposure for the period. Data on population size were obtained from the Demo ISTAT website and were used as estimates of population size by location, which by definition are the total person-years of exposure for a calendar year. 

We used four forecasting models to predict the numbers of deaths: three exponential smoothing models (i.e., seasonal exponential smoothing, Holt–Winters additive, and Holt–Winters multiplicative models) and an auto-regressive integrated moving average (ARIMA) model. 

In the exponential smoothing methods (Brown, 1959; Holt, 1957; Winters, 1960), the predictions are weighted averages of past observations, with the weights decaying exponentially as the observations get older; thus, the more recent the observation, the greater the associated weight.

Exponential smoothing models are based on a description of seasonality and/or trends in the data [29].

In the seasonal exponential smoothing method, given a time series $\left\{Y_{t}:1\leq t\leq n\right\}$, the model assumes that the exponential smoothing is as follows [30,31]:(1)$$Y_{t}=\mu_{t}+s_{p}\left(t\right)+\epsilon_{t}$$
where $\mu_{t}$ represents the time-varying mean term, $s_{p}\left(t\right)$ is the time-varying seasonal contribution, and $\epsilon_{t}$ indicates disturbances.

To better capture the seasonality, Holt [32] and Winters [33] introduced a model that extends Holt’s method by adding a trend term, $\beta_{t}t$, where $\beta_{t}$ is the time-varying slope. There are two variations to this method, which differ in the nature of the seasonal component: the additive method is preferred when the seasonal variations are roughly constant throughout the series, while the multiplicative method is preferred when the seasonal variations change proportionally to the level of the series. In the additive method, the seasonal component is expressed in absolute terms in the scale of the observed series, and within each year the seasonal component will add up to approximately zero. Thus, Equation (1) becomes:(2)$$Y_{t}=\mu_{t}+\beta_{t}t+s_{p}\left(t\right)+\epsilon_{t}$$

With the multiplicative method, the seasonal component is expressed in relative terms (percentages), and the series is seasonally adjusted by dividing by the seasonal component. Within each year, the seasonal component will sum to approximately m, where m is the frequency of the seasonality. In this method, the equation is as follows:(3)$$Y_{t}={(\mu }_{t}+\beta_{t}t)s_{p}\left(t\right)+\epsilon_{t}$$

The ARIMA (auto-regressive integrated moving average) model is a particular method of analysis that is widely used in time-series statistics and forecasting. It can be expressed as ARIMA (p.d.q), where p indicates the number of autoregressive terms, d is the difference order needed to transform the time series in stationarity, and q is the value of the moving average. In our study, we used a seasonal ARIMA model; this model is able to capture the seasonal component that is not captured by standard methods using a multiplicative model. The general forecasting equation was as follows:(4)$$(1-B{)}^{d}(1-B^{s}{)}^{D}Y_{t}=\mu +\frac{\theta (B)\theta_{s}(B^{s})}{\phi (B)\phi_{s}(B^{s})}a_{t}$$
where $\phi_{s}(B^{s})$ is the seasonal autoregressive operator and $\theta_{s}(B^{s})$ is the seasonal moving average operator. To estimate the ARIMA model, three stages are necessary [34]. The first stage is to check the stationarity of the data by the augmented Dickey–Fuller test and the Phillips–Perron test, and plot the autocorrelation function (ACF) and partial autocorrelation function (PACF). Subsequently, the parameters were estimated using the maximum likelihood method verifying the Box–Jenkins assumption [35]. Finally, the best ARIMA model was chosen after the validation process. The “Time Series Forecasting System” tool of the SAS software was used to perform all these steps.

### 2.4. Goodness-of-Fit Measures

To compare the fit of the models and choose the final model, the following measures of model fit were used: the mean absolute percentage error (MAPE), the mean absolute error (MAE), the root mean-square error (RMSE), and the random walk R-square (RWR2).

Let $y_{t}$ be the observed value and ${\hat{y}}_{t}$ be the forecasted value. The MAPE is the sum of the individual absolute errors divided by the observed value (i.e., each period separately). It is the average of the percentage errors:(5)$$MAPE=\frac{1}{n}\sum_{t=1}^{n}|\frac{y_{t}-{\hat{y}}_{t}}{y_{t}}|\;\times \;100\%$$

The MAPE is a widely used metric; however, it divides each error individually by the observed rate, so it is biased; high errors during periods with low rates will have a significant impact on the MAPE.

The mean absolute error (MAE) is a great indicator for measuring accuracy. As the name suggests, it is the mean of the absolute error and protects outliers:(6)$$MAE=\frac{1}{n}\sum_{t=1}^{n}|y_{t}-{\hat{y}}_{t}|$$

The RMSE is defined as the square root of the average squared error; it gives greater importance to the highest errors, but it ensures an unbiased forecast:(7)$$RMSE=\sqrt{\frac{1}{n}\sum_{t=1}^{n}(y_{t}-{\hat{y}}_{t}{)}^{2}}$$

The R-squared value measures the amount by which the error variance of the regression model is lower than that of the mean model for purposes of predicting the dependent variable. However, if the mean model is not an appropriate reference point, this is a meaningless statistic. It is more appropriate to use the random walk model as a reference; this model assumes that in each period the variable takes a random step away from its previous value, and that the steps are independently and identically distributed in size (“i.i.d.”). Thus, the random walk R^2^ statistic (R^2^ statistic using the random walk model for comparison) is more appropriate for forecasting models than the simple R^2^ statistic.
(8)$${RWR}^{2}=1-\left(\frac{n-1}{n}\right)\frac{SSE}{RWSSE}$$
where $µ=\frac{1}{n-1}\sum_{t=2}^{n}y_{t}-y_{t-1}$ and $RWSSE=\sum_{t=2}^{n}\left(y_{t}-y_{t-1}-µ\right)$.

### 2.5. Software

Data management, descriptive statistics, and forecasting models were performed using SAS/STAT version 9.4 for PC (SAS Institute, Cary, NC, USA).

## 3. Results

In the province of Taranto, over the period spanning from 2011 to 2021, 3108 deaths were recorded with an indication of lung cancer as the main cause or amongst other causes. The annual number of deaths decreased from 2011 (*n* = 318, rate = 54 per 10,000) to 2016 (*n* = 230, rate = 39.7 per 10,000); in 2017, 293 deaths were recorded with a rate of 50.9 per 10,000, remaining approximately stable until 2021. In the municipality of Taranto, 1279 total deaths from lung cancer were recorded, distributed from 2011 to 2021, as shown in Table 1, which also shows the main characteristics of the deaths recorded by the ReMo. The deaths were mainly in male subjects, with a percentage that varied in the entire province between 76.8% in 2017 and 85.8% in 2012, while in the municipality of Taranto the rate of male deaths decreased over the explored period, reaching its lowest value of 71.2% in 2021. Deaths in the younger age group (<66 years) decreased over time, while they increased in the older age group (>88 years), both in the province and municipality of Taranto.

In our database, 92.2% of records were identified with the ICD10 codes for lung cancer in the death certificate as the main cause of death (in only the municipality of Taranto it was 91.6%). The most frequent cause of death was “Malignant neoplasms of independent (primary) multiple sites” when the lung cancer code was not the first cause among deaths from non-primary lung cancer (Table 2). Comorbidities associated with deaths where lung cancer was the main cause of death or secondary diagnosis are shown in Appendix A.

The 2011–2021 monthly time series of mortality rates indirectly standardized by gender and age group, using the average monthly mortality rates recorded in Apulia in the five-year period of 2015–2019 as a reference, are shown in Figure 1, both for the entire province and for the municipality of Taranto.

The SMR was almost always higher than unity, both in the municipality of Taranto and in the entire province, confirming a greater risk of death from lung cancer in the area under examination compared to the regional reference. The mortality exhibited a downward trend until mid-2016, and then it rose again until mid-2019, before tending to decrease once more until the end of the study period. SMRs were higher in the municipality of Taranto than in the wider province, and a greater variability was observed in the municipality. The series of adjusted mortality rates followed the same trend as the SMRs (Appendix A).

The adjusted mortality rate series were truncated in February 2020—before the start of the pandemic—and four forecasting models were applied to them: seasonal exponential smoothing, the Holt–Winters additive method, the Holt–Winters multiplicative method, and ARIMA (2,0,0) (1,0,0)s.

The goodness-of-fit measures were compared in order to choose the best model. The Holt–Winters additive model was chosen both for the series of provincial rates and for those of the single municipality of Taranto, as it displayed lower RMSEs (0.817 and 1.542 for the province and municipality of Taranto, respectively) and MAPEs (17.37 and 29.20 for the province and municipality of Taranto, respectively) and higher RWR2s (0.35 and 0.42 for the province and municipality of Taranto, respectively) when compared with the other models. The lowest MAE was from the Holt–Winters additive model for the province (0.67) and from the ARIMA model for the municipality of Taranto (1.24) (Table 3).

The mortality rates expected from March 2020 to December 2021 in the absence of the event that may have changed the historical series (i.e., the COVID-19 pandemic) were estimated using the Holt–Winters additive model. The historical series of rates estimated with the forecasting model, along with their 95% confidence intervals and the standardized mortality rates observed before and during the pandemic, are shown in Figure 2a,b for the province and the municipality of Taranto, respectively.

In the province of Taranto, almost all of the adjusted monthly mortality rates observed during the pandemic fell within the confidence interval of the predicted rates, with the exception of the rates observed in March and August 2020, where statistically significant excess rates were recorded: 1.82 (95% CI 0.11–3.08) and 2.09 (95% CI 0.20–3.44), respectively. In the municipality of Taranto, the observed rate in August 2020 was the only one significantly higher than the predicted rate, at 3.51 (95% CI 0.33–6.69). The number of excess deaths related to COVID-19 was calculated by multiplying the expected excess death rate by the person-years of exposure for the period. The summarized quarterly results are shown in Table 4.

The highest excesses of lung cancer deaths were recorded between March and September 2020, i.e., during the total lockdown and the months immediately following, and in the period of April–June 2021. However, the excess deaths were not statistically significant in any pandemic period in either the province or the municipality of Taranto.

## 4. Discussion

In this study, we wanted to examine whether the spread of COVID-19 and the public health measures adopted to contain it had any effects on the rate of deaths from lung cancer. To the best of our knowledge, this is the first study evaluating the effects of the COVID-19 pandemic on lung cancer in a complex setting with environmental pollution.

The increased environmental risk could definitely be a confounder for our outcomes, but we strongly believe that this represents a major opportunity to assess and to debate its weight in areas at similar levels of environmental risk. Although the principal cause of lung cancer is cigarette smoke, studies have underlined the effects of pollution on this illness [36,37,38]. It has been demonstrated that some molecules present in airborne pollutants, such as PM2.5, PM10, SO2, and O3, can cause lung cancer in both urban and rural areas in the United States and China [39]. Furthermore, it is important for all states to reduce chemical waste, the byproducts of industrial and agricultural facilities, and the levels of urban traffic in order to prevent the development of this type of cancer.

In a previous report from Italy, Rocco et al. from the Naples area retrospectively analyzed the 5-year survival rates of lung cancer patients originating from a well-known polluted area known as the “Land of fire” compared to patients from other areas with no environmental issues. Their analyses demonstrated that surgical candidates had similar long-term survival and that originating from a polluted area did not seem to be associated with worse outcomes after surgical treatment of patients with lung cancer [40]. Another Italy-based study carried out in Porto Marghera—an area at high environmental risk due to industrial pollution—showed an excess in the overall cancer mortality rate in both genders, as well as in the lung cancer incidence and mortality rates. In addition, an increase in mortality from respiratory diseases was detected only in the male population [41]. Taranto is also a polluted place because there are large-scale industrial activities near the city, including steel production (one of the biggest in Europe), mineral storage, cement production, fuel storage and production, waste material management, and mining. The effects of pollution in this area have been investigated by numerous researchers. Every study has underlined the role of the polluting facilities in affecting the quality of life, with increases in the incidence of oncological illness (especially of the trachea, bronchi, and lungs) and of chronic diseases (especially cardiovascular and respiratory diseases) [5,7,42,43]; these conditions could have contributed to excess mortality rates from lung cancer with respect to the regional average.

To determine excess mortality, it is necessary to estimate mortality in a reference period—usually a historical series of the previous 5 years. We used a longer period as a reference to take into account a possible trend component, as the geographical area under study has been affected in recent years by legislative interventions aimed at mitigating the environmental risks. Therefore, we used the consolidated monthly mortality data for the previous 9 years (instead of 5), including the months of January and February 2020 (before the pandemic started), and the reference was determined based on a longer series before the pandemic started.

Although the area is at high risk for death, the observed number of cases was small, even considering that we performed our analysis on a small area. In this regard, several authors have estimated the excess mortality by using an overdispersed Poisson model that accounts for temporal trends and seasonal variability in mortality [44,45] or a zero-inflated Poisson autoregressive model and zero-inflated negative binomial autoregressive model [46]. The municipality of Taranto and its nearby areas already have an excess of deaths from lung cancer given their environmental exposure [6,8], but this condition is lost using the count of deaths as an outcome, even when adjusted for sex and age. Therefore, we used the standardized and adjusted rates with respect to a reference geographical area. The estimation method for forecasting was chosen between exponential smoothing and ARIMA. The exponential smoothing technique is widely used for forecasting, and it is one of the most popular forecasting methods for short-term periods [47,48,49]. ARIMA models have also been used by several authors to estimate excess mortality [50,51,52].

The forecasting metrics were consistent in indicating the Holt–Winters additive exponential model as more accurate, and they were worse for the series of rates in the municipality of Taranto given the smaller population at risk and therefore the lower number of deaths. Regardless of the model used, comparable measures of GoF (such as MAPE) were worse than the predictions made for non-specific causes of death or for larger geographical aggregations (such as the Apulia region or Italy) [53]. However, an MAPE value between 10 and 20 for the series in the province of Taranto (17.4) indicates a “good forecasting” according to the thresholds proposed by Lewis [54]. Meanwhile, for the series in the municipality of Taranto, the forecasting was “reasonable” given the MAPE of 29.2. This value indicates a level of uncertainty that is reflected in a wide confidence interval in the forecasting, which could have hidden other possible monthly excesses of mortality in the municipality of Taranto.

It is important to note that our model estimates expected deaths without the presence of COVID-19 infection. Subsequently, to evaluate the possible direct and indirect effects of the pandemic on the general state of health of lung cancer patients, we compared the cases expected by the model with those observed. In addition, rates were adjusted using the whole regional rate for Apulia as a standard reference.

On the one hand, the public health measures adopted by several states around the world for the containment of COVID-19 reduced the numbers of people infected by the virus, but on the other hand, they reduced access to health services and caused increases in mortality [55,56,57,58,59,60,61]. In particular, some international scientific societies recommended reducing the administration of antitumor therapy, surgery, and radiotherapy; consequently, surgical procedures were postponed and screening exams were delayed [62,63,64]. Considering the characteristics of people with lung cancer, we expected a higher mortality rate in this cohort because they have a predisposition to severe infection. Furthermore, when patients are diagnosed with lung cancer, they are often already at an advanced stage with metastases [65].

However, our study showed no significant difference in the adjusted mortality rates after the diffusion of COVID-19, except in the months of March and August 2020 for the entire province of Taranto, and only for month of August for the municipality of Taranto. In March, the pandemic become relevant in Italy in terms of the spread of COVID-19; for this reason, the government implemented a lockdown on all activities and reduced access to the national health services. We hypothesized, as proposed in other studies [56,66], that the reduced access to health services for both oncological therapy and acute diseases (e.g., cardiovascular disease) could have caused excess deaths. After the lockdown measures, we observed an increase in the spread of COVID-19 in the following months, with a relative increase in the incidence rates in Apulia and in Italy as a whole. For this reason, some restriction measures were applied once more in August. It is important to remember that the province of Taranto is also affected by a high index of socioeconomic deprivation, which could have played an important role in the spread of SARS-CoV-2 [67]. In the remainder of the study period, we observed an absence of excess mortality both in the province of Taranto and in its municipality, probably due to the strategies applied by the Regional Center for Oncological Orientation in S.G. Moscati Hospital (Taranto, Italy) [68]. During the pandemic, this center, with the help of the local and regional health authorities, aimed to minimize any interruption of oncological treatment by reducing the workload on the public facilities and reducing the waiting lists for instrumental clinical examinations. For these reasons, the oncological network was extended by introduction of 11 private hospital sites in the province of Taranto. This action was indispensable for increasing the number of prime oncological visits, completing preparations for therapy, and activating exemptions for pathology (i.e., free care) and instrumental investigations.

Our study found a generally lower trend of lung cancer deaths in the province of Taranto and in its municipality from 2011 to 2020, although there have been many fluctuations over the years. This reduction is consistent with the findings of other studies around the world. For example, in America, a reduction was found in the mortality rate of non-small-cell lung cancer (NSCLC). This reduction was caused by the application of target therapy since 2013 for stage IV EGFR-positive NSCLC [69].

However, the number of deaths in the cohort aged over 65 years increased in the same period. Multiple causes could explain this observation, including better clinical management of cancer patients both in hospitals and in home care, the awareness campaigns conducted by health services, and a possible reduction in pollution through the application of environmental measures.

This research focused only on death from a specific cancer, i.e., lung cancer, but a recent study conducted in the municipality of Taranto has shown an increase in all-cause mortality rates from 2011 to 2020, especially for the areas situated near industrial activity (e.g., Paolo VI, Tamburi, Lido Azzurro, and Old Town) [43].

Our study has some limitations. First, the use of administrative data could affect the results because they depend on disease coding and therefore the quality of the data could be dependent on the operators. Furthermore, the data are consolidated and unmodifiable; the consequence could be a bias (i.e., underestimation or overestimation) in the counts of cases and rates. Second, we explored only the excess deaths in a small area at environmental risk; although we applied standardization with respect to the whole region of Apulia, we did not compare models built on ReMo data for areas with the same population but without environmental risk. Such a comparison could support the observation that the excess mortality was not further increased by COVID-19.

## 5. Conclusions

In summary, our analysis shows that there was no excess mortality from lung cancer as a result of the COVID-19 pandemic in the province of Taranto. We hypothesize that the absence of this effect was due to the excess all-cause mortality already present in the province of Taranto compared to the wider Apulia region, because there are more sources of pollution in this small area. Furthermore, the strategies applied by the Regional Center for Oncological Orientation during the pandemic could have minimized the possible interruption of cancer treatments by reducing the workload on the public facilities and reducing the waiting lists for instrumental clinical examinations.

We also believe that it is necessary to continue to conduct studies in the city of Taranto and in its province to verify whether an excess of all-cause mortality was present after the spread of COVID-19.

## Figures and Tables

**Figure 1 ijerph-20-05522-f001:**
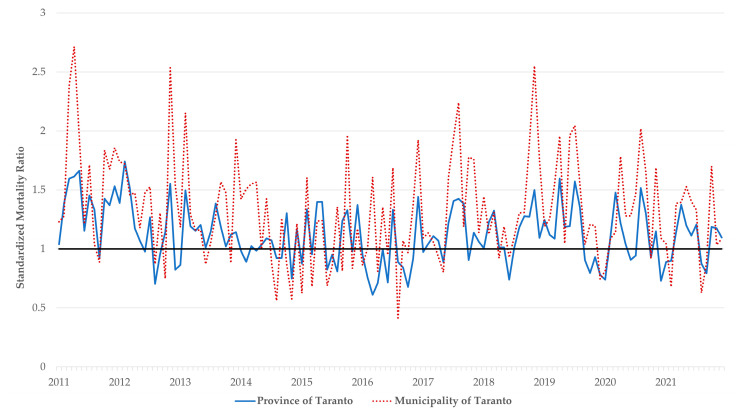
Time series of standardized mortality ratios for the province and the municipality of Taranto in the period 2011–2021.

**Figure 2 ijerph-20-05522-f002:**
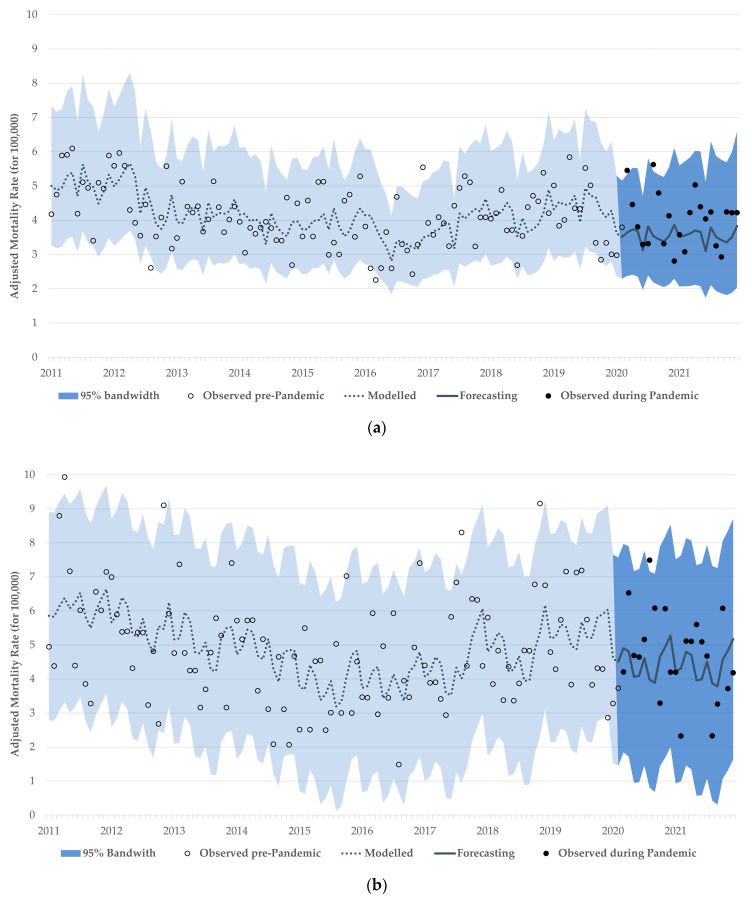
Time series of AMRs and the forecasting model estimated with the Holt–Winters additive method: (**a**) province of Taranto and (**b**) municipality of Taranto.

**Table 1 ijerph-20-05522-t001:** Main characteristics of lung cancer deaths recorded in the province of Taranto and in the municipality of Taranto between 2011 and 2021.

Geographic Area	Year	Deaths for Lung Cancer	Crude Rate (per 10,000)	Lung Cancer as the Main Cause	Sex (Male)	Age			
						<66 Years	66–75 Years	76–85 Years	≥86 Years
Province of Taranto *	2011	318	54.0	293 (92.1)	260 (81.8)	89 (28)	106 (33.3)	98 (30.8)	25 (7.9)
	2012	281	47.7	261 (92.9)	241 (85.8)	78 (27.8)	97 (34.5)	86 (30.6)	20 (7.1)
	2013	278	47.4	257 (93.6)	225 (80.9)	70 (25.2)	99 (35.6)	93 (33.5)	16 (5.8)
	2014	248	42.4	232 (93.6)	201 (81.1)	61 (24.6)	95 (38.3)	70 (28.2)	22 (8.9)
	2015	280	48.1	257 (91.8)	224 (80)	55 (19.6)	103 (36.8)	94 (33.6)	28 (10)
	2016	230	39.7	221 (96.1)	186 (80.9)	47 (20.4)	90 (39.1)	67 (29.1)	26 (11.3)
	2017	293	50.9	276 (94.3)	225 (76.8)	52 (17.8)	99 (33.8)	113 (38.6)	29 (9.9)
	2018	297	51.9	268 (90.2)	250 (84.2)	66 (22.3)	96 (32.3)	103 (34.7)	32 (10.8)
	2019	302	53.1	278 (92.1)	243 (80.5)	59 (19.5)	100 (33.1)	101 (33.4)	42 (13.9)
	2020	289	51.2	265 (91.7)	231 (79.9)	53 (18.3)	102 (35.3)	109 (37.7)	25 (8.7)
	2021	292	52.0	259 (88.7)	233 (79.8)	45 (15.4)	108 (37)	104 (35.6)	35 (12)
	Total	3108	48.9	2867 (92.2)	2519 (81)	675 (21.7)	1095 (35)	1038 (33.4)	300 (9.7)
Municipality of Taranto	2011	132	65.8	122 (92.4)	104 (78.8)	43 (32.6)	51 (38.6)	24 (18.2)	14 (10.6)
	2012	120	59.9	110 (91.7)	97 (80.8)	37 (30.8)	43 (35.8)	35 (29.2)	5 (4.2)
	2013	111	55.7	99 (89.2)	89 (80.2)	31 (27.9)	38 (34.2)	32 (28.8)	10 (9)
	2014	98	49.3	89 (90.8)	83 (84.7)	24 (24.5)	28 (28.6)	32 (32.7)	14 (14.3)
	2015	95	48.3	87 (91.6)	79 (83.2)	20 (21.1)	33 (34.7)	34 (35.8)	8 (8.4)
	2016	104	53.2	99 (95.2)	78 (75)	23 (22.1)	46 (44.2)	24 (23.1)	11 (10.6)
	2017	125	64.4	120 (96)	90 (72)	23 (18.4)	40 (32)	49 (39.2)	13 (10.4)
	2018	128	66.4	113 (88.3)	101 (78.9)	26 (20.3)	40 (31.3)	45 (35.2)	17 (13.3)
	2019	128	67.0	117 (91.4)	97 (75.8)	26 (20.3)	44 (34.4)	41 (32)	17 (13.3)
	2020	127	66.6	118 (92.9)	95 (74.8)	23 (18.1)	49 (38.6)	46 (36.2)	9 (7.1)
	2021	111	58.7	98 (88.3)	79 (71.2)	19 (17.1)	45 (40.5)	37 (33.3)	10 (9)
	Total	1279	676.1	1172 (91.6)	992 (77.6)	295 (23.1)	457 (35.7)	399 (31.2)	128 (10)

Data shown as frequency (percentage). * includes the municipality of Taranto.

**Table 2 ijerph-20-05522-t002:** Counts and percentages of main causes of death in records where lung cancer was a secondary diagnosis.

Main Cause of Death		*n* (%)
ICD10 Code	Label	
C97	Malignant neoplasms of independent (primary) multiple sites	27 (11.2)
J44.8	Other specified chronic obstructive pulmonary disease	12 (5)
I11.9	Hypertensive heart disease without (congestive) heart failure	10 (4.1)
I25.9	Chronic ischaemic heart disease, unspecified	10 (4.1)
E14.9	Unspecified diabetes mellitus: Without complications	9 (3.7)
U07.1	COVID-19	8 (3.3)
C61	Malignant neoplasm of prostate	7 (2.9)
C67.9	Malignant neoplasm of bladder, unspecified	6 (2.5)
J44.9	Chronic obstructive pulmonary disease, unspecified	6 (2.5)
	Other diseases	146 (60.6)
	Total	241

**Table 3 ijerph-20-05522-t003:** Measures of goodness of fit of forecasting models applied to the series of adjusted mortality rates.

Model	Province of Taranto				Municipality of Taranto			
	RMSE	MAE	MAPE	RWR^2^	RMSE	MAE	MAPE	RWR^2^
Seasonal Exponential Smoothing	0.819	0.68	17.58	0.34	1.545	1.26	29.68	0.41
Winters Method—Additive	0.817	0.67	17.37	0.35	1.542	1.25	29.20	0.42
Winters Method—Multiplicative	0.855	0.69	17.94	0.28	1.615	1.31	29.93	0.36
ARIMA (2,0,0) (1,0,0)s	0.831	0.69	17.73	0.32	1.561	1.24	29.22	0.40

The best model is highlighted in gray.

**Table 4 ijerph-20-05522-t004:** Estimated excess lung cancer deaths during the pandemic and their 95% confidence intervals in the province and the municipality of Taranto.

Pandemic Period	Province of Taranto	Municipality of Taranto
March 2020–June 2020	+16 (−23; +45)	+4 (−19; +28)
July 2020–September 2020	+17 (−16; +40)	+12 (−6; +30)
October 2020–December 2020	−3 (−38; +21)	−3 (−21; +16)
Total 2020	30 (−77; 106)	14 (−47; 74)
January 2021–March 2021	+1 (−35; +26)	−3 (−22; +16)
April 2021–June 2021	+17 (−21; +42)	+6 (−13; +25)
July 2021–September 2021	−2 (−42; +25)	-4 (−23; +16)
October 2021–December 202	+11 (−31; +39)	−1 (−21; +19)
Total 2021	28 (−130; 133)	−2 (−80; 76)

## Data Availability

No new data were created or analyzed in this study. Data sharing is not applicable to this article.

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
