# Peer review of "No Excess of Mortality from Lung Cancer during the COVID-19 Pandemic in an Area at Environmental Risk: Results of an Explorative Analysis"

_ijerph, 2023, doi:10.3390/ijerph20085522_

Round 1
Reviewer 1 Report
The topic is relevant, however, the way it was presented presents many problems. The title leads the reader to believe that there has been an increase in mortality from lung cancer during the pandemic. In fact, the absolute number of deaths, which was 293 (92.1% of the total) in 2011, dropped to: 259 (88.7% of the total in 2021) and 265 (91,7% of the total in 2020). In addition, lung cancer mortality rates for the period studied were not presented. Not even if there was a significant difference between the rate calculated (according to the rates expected) and the one actually verified. Regarding the absolute numbers, the authors themselves describe that: “However, the excess deaths were not statistically significant in any pandemic period in either the province or the municipality of Taranto”. If there is no statistical difference, what high rates are the authors referring to? In fact, the authors themselves describe that for the province of Taranto the adjusted monthly rates were within the confidence interval (except in two months) and only one month if the municipality of Taranto is considered.
We have to be very careful in this post-pandemic moment, due to a large number of denialists and people who criticize restrictive measures. However, in the case of this particular article, the title and conclusions make no sense. The article leads the reader to believe that there was an excess mortality due to lung cancer (which did not happen) and states that this may be related to the restrictive measures adopted without showing any evidence that the restrictive measures harmed the patients undergoing treatment.
Title: “Excess of mortality from lung cancer during the covid-19 pandemic”. The title needs to be corrected since there was no excess mortality due to lung cancer during the period studied. So this title makes no sense at all.
The abstract needs to be improved in many points:
a) “Areas with high levels of air pollution already have a high risk of death from cancer”. Is this the reality in Taranto Province? Clearly show this relationship. Only in the introduction is this relationship clarified (related to the presence of one of the largest steel plants in the EU, a major oil refinery, a concrete factory, and a naval military base). This relationship is not known by researchers from other continents.
b) “we aimed to evaluate the effects of 20 the pandemic on excess mortality from lung cancer” At this point, I believe that the authors should clearly show the over-mortality rate (if this is known information) and in this case, the objective would be to investigate whether there is a relationship with the restrictive measures adopted. Or if the over-mortality was a result of this study, the objective should be adequate.
c) “In Taranto Province, 3108 deaths from lung cancer were recorded between 2011 and 2021” “The trend was decreasing from 2011 (n = 318) until 2016 (n =330).” I propose to use the lung cancer mortality rate, probably the population has increased, so even though the absolute number is higher, the rate is lower. As the information is presented, it is not possible to verify the downward trend. Actually verified after reading the paper that it is a mistake, the correct number for 2016 would be 230 (rate dropped from 54.0/10000 to 39.7/10000) for total deaths
and not for lung cancer as the main cause.
d) “The trend was decreasing from 2011 until 2016”. So the trend between 2016 and 2021 was already rising? In this case, what is the pre-pandemic (until 2019) and during the pandemic (2020-2021) rate?
e) “In the province of Taranto, almost all of the adjusted monthly mortality rates during the pandemic were within the confidence interval of the predicted rates, with the exception of significant excesses in March and August 2020. In the municipality of Taranto, the only significant excess rate was in August 2020”. With these results, did the variation in these months affect the expected annual rate? Did an over-mortality actually occur? What is the expected annual rate pre and during the pandemic? How much was the annual over-mortality? For example, in my country, these are the months when people return from vacation, so both March and August are always influenced by people returning for medical procedures. And this seasonal feature is unrelated to the pandemic.
f) “This study shows the trend in excess mortality for lung cancer”. I do not agree. The reading up to this point showed only one over-mortality in two specific months. I still have doubts if, in fact, considering the whole year, if there really was an over-mortality.
Introduction:
a) “Pandemic health policies have had a tremendous impact on patients with cancer.
Early in the pandemic, a WHO survey found that health services for non-communicable
diseases in many countries were partially or completely disrupted, which was later confirmed and systematically reviewed in several studies”. This has indeed been the case in many countries, but how exactly have services in Taranto been affected? Were they interrupted? If yes, for how long? Has the number of queries dropped? We are dealing with a specific epidemiological reality here. To relate a cause and a consequence, it is necessary to clarify how this occurred in the studied place. This context is more important than the review presented between lines 85 and 95.
“Preliminary raw data analyses from our group in Taranto’s Local Health Authority (unpublished data) have suggested that patients’ visits to treatment facilities have been curtailed” - What is the estimated percentage? “and treatments have, in some cases, been withheld” - What is the estimated percentage?
Materials and Methods:
a) Authorization from the ethics committee was not presented to carry out the study. As these are not public data, I would like authorization to carry out the study to be presented in the complementary data.
b) At some points in the result, the authors refer to the average number of deaths per year (as in line 234, for example). What kind of test to verify the normality of the data was applied?
c) What kind of test was applied to verify the annual differences between mortality rates?
Results
General: Authors mention data not presented in their tables or supplementary tables (such as absolute numbers, averages, etc...).
a) Please, rebuild table 1 including the lung cancer mortality rate and other data related to lung cancer mortality (and not general mortality).
b) Please correct the legend, as the data presented by age were for total mortality (not for lung cancer mortality).
c) Please, correct line 232 (n=230).
d) In terms of the mortality rate, 2016 was very much outside the curve and should not be highlighted in the analysis. It would be more interesting to verify if there is a statistical difference between the annual rates in the analyzed period.
e) “Taranto, 1279 deaths from lung cancer were recorded, with an average of ~116 deaths per year, as shown in Table 1)” - Table 1 does not show these numbers. Include the totals and the mean or median (depending on whether the data have a normal or non-normal distribution) in the table.
f) “In our database, 92.4% of records were identified with the ICD10 codes for lung cancer in the death certificate as the main cause of death”. This is true for the municipality of Taranto, but is different for the province (the data is 92.1%).
g) Table 3 - as we are working with time series, please present the estimate not by quarter but by year. You can put this information (for shorter periods) in the appendix.
h) “However, the excess deaths were not statistically significant in any pandemic period in either the province or the municipality of Taranto”. This fact should be considered when giving the paper's title and be highlighted in the abstract and conclusions. In my opinion, based on the figures shown by the authors, there was no excess mortality due to lung cancer.
Discussion
a) It should be rewritten after corrections in the presentation of results.
b) “Therefore, we could hypothesize direct and indirect effects of the pandemic on the general state of health of lung cancer patients”. - this must be done very carefully. And I don't think the strong claims in this article support the conclusions being presented.
c) Lines – 378-386- When we say that the measures to contain COVID-19 are causing increases in mortality, this must be done carefully and clearly showing the numbers and what disease we are talking about. This did not happen in this article. The measures adopted were not described, and the number of consultations, exams, and surgeries that were not performed for lung cancer was not shown. The study did not show whether there is a statistical difference in the lung cancer mortality rate during the year 2020 compared to other years. It only indicates months in which this mortality was higher.
d) Lines 393-395 - “We hypothesized, as proposed in other studies [49,54], that reduced access to health services for both oncological therapy and acute diseases (e.g., cardiovascular disease) could have caused excess deaths”. - And it seems that the article was described with this objective even before the analysis of the data. In my opinion, the precautions that must be taken in the declarations referring to excess mortality in this period, are being carried out without due reservation.
e) Lines 404-410 - I believe that this effort has had results due to the mortality rates presented in the study. Therefore, the title of this article is inappropriate. It appears that there has been an increase in the number of deaths from lung cancer, when in fact, this has not occurred.
Conclusions:
“In summary, our analysis shows that there was no excess mortality from lung cancer as a result of the COVID-19 pandemic in the province of Taranto”. In fact, the study does not show excess mortality from lung cancer, whether from any cause.
The previous statement made by the authors contrasts with the one described in the abstract: “This study shows the trend in excess mortality for lung cancer, and its findings could be
related to the implementation of lockdown policies and multiple direct and indirect pathways associated with mortality risk.” - There is not the slightest description of how the blockade measures effectively reduced patients' access to their treatment. This is even refuted in lines 400-410.
Reviewer 2 Report
The introduction and especially the discussion part need to be improved.
The theory about the ARIMA is not complete.
There are three steps for model building of ARIMA: (1) Identification (stationarity, etc); (2) Model Estimation; and (3) Diagnostic checking of Model.
But the article does not explain the steps of ARIMA in their results.
I suggest to read the references: Ahmar, et. al. (2022). Forecasting the Value of Oil and Gas Exports in Indonesia using ARIMA Box-Jenkins. JINAV: Journal of Information and Visualization
Author Response
We thank the reviewer for his really valuable comments. We have modified the text trying to follow the reviewer's suggestions. Thanks to the suggested changes, we believe that the text has improved and we hope that this revised version of the manuscript will meet the approval of the reviewer, the editor and therefore the readers. Thank you.
The introduction and especially the discussion part need to be improved.
R.
Following the reviewer's suggestion we have modified both the introduction and the discussion.
The theory about the ARIMA is not complete.
There are three steps for model building of ARIMA: (1) Identification (stationarity, etc); (2) Model Estimation; and (3) Diagnostic checking of Model.
But the article does not explain the steps of ARIMA in their results.
I suggest to read the references:
Ahmar, et. al. (2022). Forecasting the Value of Oil and Gas Exports in Indonesia using ARIMA Box-Jenkins. JINAV: Journal of Inforà.mation and Visualization
R.
We thank the reviewer for the suggestion.
We have added in the "Forecasting models" section of the methods, on line 202, the three steps indicated by the reviewer which led to the ARIMA(2,0,0)(1,0,0)s model
Reviewer 3 Report
This manuscript presents the impact of COVID-19 pandemic on death of lung cancer in the "high risk area". It provide the important findings to understand the impact of COVID-19 pandemic on public health. I have several minor comments.
#1. Although the authors have describe the period of lockdown in discussion part, it is better to describe the period of pandemic waves and lockdown in order to understand the infection situation of in Italy in introduction.
#2. I could not understand whether the authors were trying to clarify whether the COVID-19 pandemic had reduced patient access to medical institutions or whether excess mortality due to COVID-19 would affect the lung cancer mortality rate. What did you try to clarify by looking at the impact of COVID19 on lung cancer deaths? The authors should describe it in more detail and clearly.
Author Response
This manuscript presents the impact of COVID-19 pandemic on death of lung cancer in the "high risk area". It provide the important findings to understand the impact of COVID-19 pandemic on public health. I have several minor comments.
R.
We thank the reviewer for his really valuable comments. We have modified the text trying to follow the reviewer's suggestions. Thanks to the suggested changes, we believe that the text has improved and we hope that this revised version of the manuscript will meet the approval of the reviewer, the editor and therefore the readers. Thank you.
#1. Although the authors have describe the period of lockdown in discussion part, it is better to describe the period of pandemic waves and lockdown in order to understand the infection situation of in Italy in introduction.
R.
Thanks for your suggestion. We added in the introduction (at line 73) the description of the pandemic wave and the public health measures adopted in that period by the Italian government.
#2. I could not understand whether the authors were trying to clarify whether the COVID-19 pandemic had reduced patient access to medical institutions or whether excess mortality due to COVID-19 would affect the lung cancer mortality rate. What did you try to clarify by looking at the impact of COVID19 on lung cancer deaths? The authors should describe it in more detail and clearly.
R.
We thank the reviewer who allows us to clarify that our objective was to verify whether in an area at environmental risk where the mortality rate from lung cancer is already high, there may have been a further excess of mortality due to restrictions on access to public health services during the pandemic.
We clarified the aim in the introduction to lines 134-135.
Round 2
Reviewer 1 Report
I congratulate the authors for reviewing the article. I think the article is clearer and several points of confusion have been cleared up. I only request the correction of table 1. There is a typo in the title: “deaths for lung cancer”; was typed: “lnng”.